# Multifunctionality of silver *closo*-boranes

Mark Paskevicius[1,2], Bjarne R.S. Hansen[1,*], Mathias Jørgensen[1,*], Bo Richter[1] & Torben R. Jensen[1]

Silver compounds share a rich history in technical applications including photography, catalysis, photocatalysis, cloud seeding and as antimicrobial agents. Here we present a class of silver compounds ($Ag_2B_{10}H_{10}$ and $Ag_2B_{12}H_{12}$) that are semiconductors with a bandgap at 2.3 eV in the green visible light spectrum. The silver boranes have extremely high ion conductivity and dynamic-anion facilitated $Ag^+$ migration is suggested based on the structural model. The ion conductivity is enhanced more than two orders of magnitude at room temperature (up to 3.2 mS cm$^{-1}$) by substitution with AgI to form new compounds. Furthermore, the *closo*-boranes show extremely fast silver nano-filament growth when excited by electrons during transmission electron microscope investigations. Ag nano-filaments can also be reabsorbed back into $Ag_2B_{12}H_{12}$. These interesting properties demonstrate the multifunctionality of silver *closo*-boranes and open up avenues in a wide range of fields including photocatalysis, solid state ionics and nano-wire production.

[1] Department of Chemistry and Interdisciplinary Nanoscience Center (iNANO), Aarhus University, Langelandsgade 140, Aarhus C DK-8000, Denmark. [2] Department of Physics and Astronomy and Fuels and Energy Technology Institute, Curtin University, Wark Ave, Bentley Western Australia 6102, Australia. * These authors contributed equally to this work. Correspondence and requests for materials should be addressed to M.P. (email: m.paskevicius@curtin.edu.au) or to T.R.J. (email: trj@chem.au.dk).

Exceptional solid-state conductivity was first noted by Faraday in 1833 for $Ag_2S$. He noted that the conducting power was 'feeble' until heat was applied and it began conducting 'in the manner of a metal'[1]. Silver iodide is a classic example with the first-order structural transition at 146 °C that results in a step-function increase in $Ag^+$ conductivity, 5–6 orders of magnitude higher than at room temperature[2]. AgI is not ideal for technical applications due to poor ion conductivity at room temperature. However, a wide range of silver compounds have been investigated including $RbAg_4I_5$ (ref. 3), which displays one of the highest room temperature solid-state ion conductivities[4]. Silver batteries based on this solid-state electrolyte, $Ag|RbAg_4I_5|I_2$, show impressive performance even after 20 years of storage[5]. Recently, metal boranes have been identified as highly promising solid state ion conductors[6]. For example, $Na_2B_{10}H_{10}$ and $Na_2B_{12}H_{12}$ have shown high ion conductivities that increase to $0.01\,S\,cm^{-1}$ above a polymorphic transition temperature ($T > 110\,°C$), where the closo-borane anions undergo rapid dynamic motion and $Na^+$ ions can move between partially occupied sites in the crystal structure[7–9]. A large variety of transition metal closo-boranes have previously been synthesized (that is, Ag, Cd, Co, Cr, Cu, Fe, Hg, Mn, Ni, Pd, Sc, Zn)[6], in most cases as solvates due to the strong solvent coordination to the cation. In contrast, the silver cation has a lower charge density and can be isolated as a solvent-free closo-borane[10], but interestingly, only the crystal structures of silver closo-borane solvates have previously been described[11]. Silver compounds tend to exhibit another interesting property, the formation of silver filaments, otherwise known as whiskers, capillaries, hair, moss, fibres, wires, spikes, clusters or filiforms. Silver nanowires are interesting in their own right, with potential applications in electronics, photonics, optoelectronics, catalysis and medicine[12].

Here we present the synthesis, structures and polymorphic transitions of $Ag_2B_{10}H_{10}$ and $Ag_2B_{12}H_{12}$ along with fascinating properties such as semiconductivity, extreme room temperature ion conductivities and metallic filament growth.

## Results

**Structural dynamics.** The two silver closo-boranes, $Ag_2B_{10}H_{10}$ and $Ag_2B_{12}H_{12}$, both exhibit reversible polymorphic transitions at 180 and 200 °C, respectively, as shown in Fig. 1. The room temperature $\alpha$-$Ag_2B_{10}H_{10}$ polymorph crystallizes in a tetragonal unit cell in space group $P4/nnc$. Ag-atoms are located in the $4d$ Wyckoff site and coordinate tetrahedrally to four $B_{10}H_{10}^{2-}$ anions with an Ag − H ($\eta^1$) bond length of $\sim 2.0$ Å. The room temperature $\alpha$-$Ag_2B_{12}H_{12}$ polymorph is cubic ($Pa$-3) and isostructural to $\alpha$-$Li_2B_{12}H_{12}$ (ref. 13) and $Ag_2B_{12}Cl_{12}$ (ref. 14). The $B_{12}H_{12}^{2-}$ anions are centred on the faces and corners of the unit cell, while $Ag^+$ occupies the $8c$ Wyckoff sites. Each Ag-cation coordinates to three $B_{12}H_{12}$-anions with Ag − H ($\eta^2$) bond lengths of $\sim 2.1$–2.4 Å. Further structural details are provided in Table 1, Supplementary Table 1 and Supplementary Figs 1–4.

The crystal structures of the high temperature polymorphs $\beta$-$Ag_2B_{10}H_{10}$ and $\beta$-$Ag_2B_{12}H_{12}$ are isostructural to one another ($Fm$-$3m$), which is clearly seen from the similarities between the high temperature diffraction patterns in Fig. 1. There is a broad and diffuse X-ray scattering background ($\sim 10° < 2\theta < \sim 18°$) that accompanies the $\beta$-polymorphs. This is indicative of rapid ionic motion due to correlations between disordered ions similar to that observed for $Ag_2S$ (ref. 15), $Li_2B_{12}H_{12}$ and $Na_2B_{12}H_{12}$ (ref. 9), due to cation disorder and/or rapid anion motion. The closo-borane anions are structurally described by reoriented partially occupied B − H polyhedra. The structural model extracted from diffraction data cannot differentiate between static or dynamic orientational disorder, but dynamic disorder is well-known in other borane systems from comprehensive studies using nuclear magnetic resonance, quasielastic neutron scattering and neutron vibrational spectroscopy[16–18].

The silver ions in the $\beta$-structures are distributed over two different sites, $32f$ and $8c$, occupying each crystallographic site equally. The $8c$ site is tetrahedrally surrounded by four $32f$ sites, as shown by the transparent tetrahedra in Fig. 1, but only one of these five sites can be occupied by Ag simultaneously ($32f$ site separation $\sim 2.5$ Å compared to the $Ag^+$ diameter of 2.58 Å). Each face of the tetrahedron is directed towards an anion superposition and the anions generate an inverted tetrahedron with respect to the Ag-tetrahedron around the $8c$ Wyckoff site. The $8c$ Ag sites permit plausible Ag − H bond lengths (2.0–2.6 Å) for all anion orientations, but are more likely in certain orientations (Supplementary Fig. 5). The $32f$ Ag sites are subjected to a large variability in Ag − H bond lengths (1.4–2.8 Å) depending on the anion orientation. In many cases the $32f$ Ag site has an Ag − H bond length that is physically unrealistic when compared to known Ag − H bond lengths (1.8–2.7 Å) (ref. 11). This suggests that Ag must occupy other crystallographic sites, when the anion is in these orientations, thus supporting the theory of dynamic-anion facilitated $Ag^+$ migration, where Ag-ions may be forced to jump from site-to-site by rotating anions.

$\beta$-$Ag_2B_{10}H_{10}$ and $\beta$-$Ag_2B_{12}H_{12}$ decompose into metallic Ag and a non-crystalline boron-rich compound above 280 and 320 °C, respectively, demonstrating reasonable thermal stability. Interestingly, $Ag_2B_{12}H_{12}$ decomposes endothermically, but $Ag_2B_{10}H_{10}$ decomposes exothermically, suggesting a different decomposition mechanism (Supplementary Figs 6 and 7).

**Ion conductivity.** Two compounds were discovered in the $AgI - Ag_2B_{10}H_{10}$ and $AgI - Ag_2B_{12}H_{12}$ systems, synthesized by annealing 1:1 compositions to study the influence of iodide substitution on ion conductivity. In situ SR-PXD of these composites reveals discontinuous changes in Bragg diffraction (Supplementary Figs 8 and 9), which suggest a fixed stoichiometry for new compounds, denoted $Ag_{(2+x)}I_xB_{10}H_{10}$ and $Ag_{(2+x)}I_xB_{12}H_{12}$, possibly with $x \sim 1$, rather than the formation of solid solutions. Other 1:1 stoichiometric compounds incorporating metal iodides and boranes have been shown previously for $M_3IB_{12}H_{12}$ ($M = NH_4$, K, Rb, Cs)[19]. The $Ag^+$ ion conductivity for $Ag_2B_{10}H_{10}$, $Ag_2B_{12}H_{12}$, $Ag_{(2+x)}I_xB_{10}H_{10}$ and $Ag_{(2+x)}I_xB_{12}H_{12}$ are shown in Fig. 2. $Ag_2B_{10}H_{10}$ and $Ag_2B_{12}H_{12}$ have similar ion conductivities and both show a step function in ion conductivity near $T = 180$ or 200 °C after the $\alpha$- to $\beta$-polymorphic transition, similar to other metal-$B_{10}H_{10}$ or -$B_{12}H_{12}$ systems. The new $Ag_{(2+x)}I_xB_{10}H_{10}$ and $Ag_{(2+x)}I_xB_{12}H_{12}$ display vastly improved room temperature ion conductivity in comparison to their parent compounds: AgI, $Ag_2B_{10}H_{10}$ and $Ag_2B_{12}H_{12}$ (Fig. 2). The mixed iodide-boranes form compounds with different structures that improve the room temperature ion conductivity by over two orders of magnitude. At room temperature, the ion conductivity of $Ag_{(2+x)}I_xB_{10}H_{10}$ and $Ag_{(2+x)}I_xB_{12}H_{12}$ are higher than any other metal borane discovered so far (Fig. 2). However, the sodium carboranes ($NaCB_9H_{10}$ and $NaCB_{11}H_{12}$) do exhibit higher ion conductivities at elevated temperature, which can be stabilized to room temperature by quenching, at least temporarily[20,21]. Iodide incorporation has also been shown to enhance the ion conductivity of $LiBH_4$ by substitution with LiI (Fig. 2) and stabilization of the high temperature polymorph to room temperature[22].

**Semiconductivity.** The silver closo-boranes, $Ag_2B_{10}H_{10}$ and $Ag_2B_{12}H_{12}$, display photosensitivity analogous to silver halides,

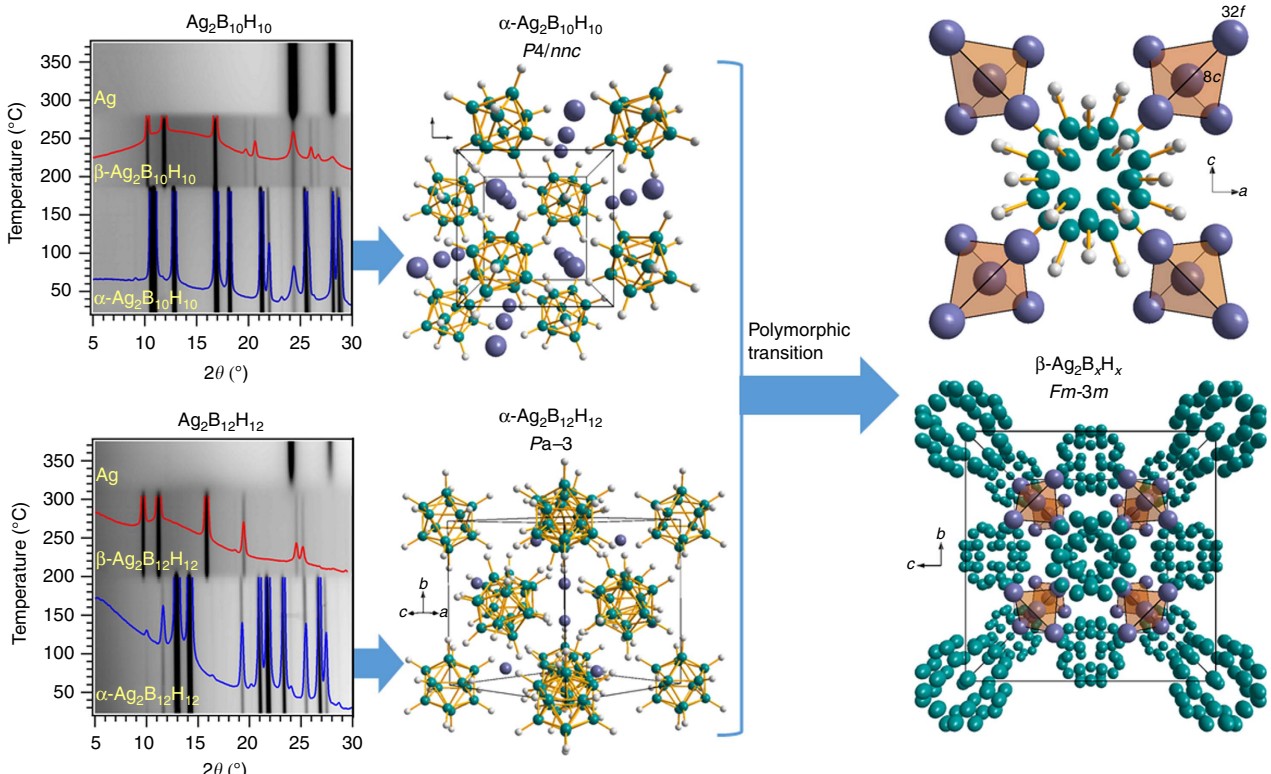

**Figure 1 | Polymorphic transformations of silver *closo*-boranes.** *In-situ* synchrotron radiation powder X-ray diffraction (SR-PXD) data during constant heating at 5 °C per min for $Ag_2B_{10}H_{10}$ and $Ag_2B_{12}H_{12}$ ($\lambda = 0.9938\,\text{Å}$). The two low temperature crystal structures ($\alpha$) are displayed along with their high temperature polymorph ($\beta$).

**Table 1 | Crystal structure data for silver *closo*-borane polymorphs.**

| Chemical formula | $\alpha$-$Ag_2B_{10}H_{10}$ | $\beta$-$Ag_2B_{10}H_{10}$ | $\alpha$-$Ag_2B_{12}H_{12}$ | $\beta$-$Ag_2B_{12}H_{12}$ |
|---|---|---|---|---|
| Crystal system | Tetragonal | Cubic | Cubic | Cubic |
| Space group | $P4/nnc$ | $Fm$-$3m$ | $Pa$-$3$ | $Fm$-$3m$ |
| $a$ (Å) | 6.29525(18) | 9.6519(4) | 9.7654(3) | 10.1395(12) |
| $c$ (Å) | 10.4330(5) | – | – | – |
| $V$ (Å³) | 413.46(3) | 899.16(6) | 931.27(5) | 1042.4(2) |
| $Z$ | 2 | 4 | 4 | 4 |
| $M$ (g mol⁻¹) | 333.93 | 333.93 | 357.56 | 357.56 |
| $\rho_{calc}$ (g ml⁻¹) | 2.683 | 2.464 | 2.531 | 2.279 |
| $T$ (°C) | 25 | 230 | 25 | 255 |

which are well-known for their photoactive properties that have enabled the development of modern photographic film[23]. During synthesis, both materials are initially white powders but become darker in colour over time due to the presence of metallic silver. In fact, the silver *closo*-boranes absorb light over a wide energy band through infrared, visible light and ultra-violet (Supplementary Fig. 10). $Ag_2B_{10}H_{10}$ and $Ag_2B_{12}H_{12}$ both exhibit bandgaps of 2.3 eV (green, 539 nm), lower than that of AgI, 2.8 eV (blue, 443 nm). Thus silver *closo*-boranes provide the possibility to be activated by a larger portion of the solar energy spectrum ($\sim 25\%$) than ultraviolet based photocatalysts ($\sim 5\%$). These photoabsorption properties may allow silver *closo*-boranes to act as visible-light photocatalysts. The silver boranes are one of the very few complex hydrides that exhibit semiconductivity, along with the first metal borohydride semiconductor, $CsPb(BH_4)_3$ (ref. 24). A series of Ag-based photocatalysts were recently identified[25], with $Ag_3PO_4$ demonstrating a quantum yield of nearly 90% (bandgap $\sim 2.4$ eV), performing well in both water splitting and waste-water cleaning applications[26]. Thus,

the silver *closo*-boranes may also show promise as future photocatalysts due to their water stability and favourable bandgap.

**Silver nano-filament growth**. The silver *closo*-boranes also undergo rapid electron-driven reduction when imaged by electron microscopy (Fig. 3a and Supplementary Movie 1). Silver filaments (10–50 nm thickness) are rapidly folded on themselves from nucleation points on the surface of the *closo*-borane particle as they are expelled into woven-like bundles of Ag fibres (Fig. 3; Supplementary Fig. 11). The quantity of silver that is expelled is far greater than the local silver content near the $Ag_2B_{12}H_{12}$ nucleation point, demonstrating excellent $Ag^+$ conductivity and long-range ion migration from other regions of the particle and possibly through interfaces from other particles. The Ag filament nucleation point appears to allow for the autocatalytic reduction of $Ag^+$: if the Ag filament folds back and touches the $Ag_2B_{12}H_{12}$ surface it acts as a second nucleation point, allowing a new

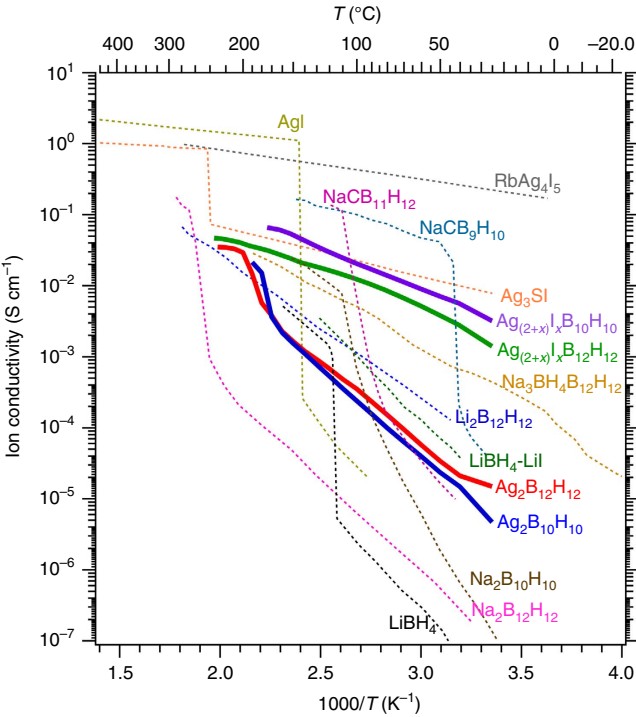

**Figure 2 | Ion conductivity of the new silver *closo*-boranes.** The data (bold) are displayed in comparison to other silver compounds (AgI, RbAg$_4$I$_5$, Ag$_3$SI)[36] and metal boranes from the literature (Na$_2$B$_{10}$H$_{10}$ (ref. 7), Li$_2$B$_{12}$H$_{12}$ (ref. 8), Na$_2$B$_{12}$H$_{12}$ (ref. 8), NaCB$_9$H$_{10}$ (ref. 21), NaCB$_{11}$H$_{12}$ (ref. 20), LiBH$_4$ (ref. 37), LiBH$_4$-LiI (ref. 22), Na$_3$BH$_4$B$_{12}$H$_{12}$)[38].

filament to form (Fig. 3d and Supplementary Movie 1). In some cases the Ag filaments are also able to be reabsorbed back into the parent Ag$_2$B$_{12}$H$_{12}$ particle (Fig. 3c; Supplementary Movie 1). Ag$_2$B$_{12}$H$_{12}$ is heavily bombarded by the electron beam during TEM imaging and the high electron current may promote the reduction reaction:

$$Ag^+ + e^- \leftrightarrow Ag \qquad (1)$$

As silver is reduced there will be a negative charge build up due to the isolated B$_{12}$H$_{12}^{2-}$ anion. However, the borane anion could also be oxidized to monovalent or neutral species, which have been theoretically and experimentally investigated (that is, B$_{12}$X$_{12}^-$ or B$_{12}$X$_{12}$, X = H, F, Cl, Br, I)[27–29].

The Ag formation mechanism may begin in a similar way to the photographic process, where silver atoms are formed at a point within a silver salt when illuminated by photons. These photons promote the formation of electron hole pairs via the photovoltaic effect and a free electron can encounter a lattice defect causing it to become negatively charged, capture Ag$^+$ ions, and form Ag (ref. 30). Similarly, free electrons in Ag$_2$B$_{12}$H$_{12}$ could promote Ag formation at defect sites. Here, Ag metal can effectively capture and conduct electrons from the charged metallic Ag filament to the Ag$_2$B$_{12}$H$_{12}$ contact point, promoting fresh Ag growth. There is limited control over the growth process within the TEM, but an increase in electron current typically initiates or enhances Ag growth, which is still localized to particular nucleation sites. Interestingly, the Ag morphology is always filament-like from these nucleation sites, where silver is drawn from distant regions of the powder. However, the electron-driven process is different to thermal decomposition, where thermally treated Ag$_2$B$_{12}$H$_{12}$ shows 50–500 nm Ag particles without filament growth (Supplementary Fig. 12). The study of nanocrystal growth mechanisms is still a developing

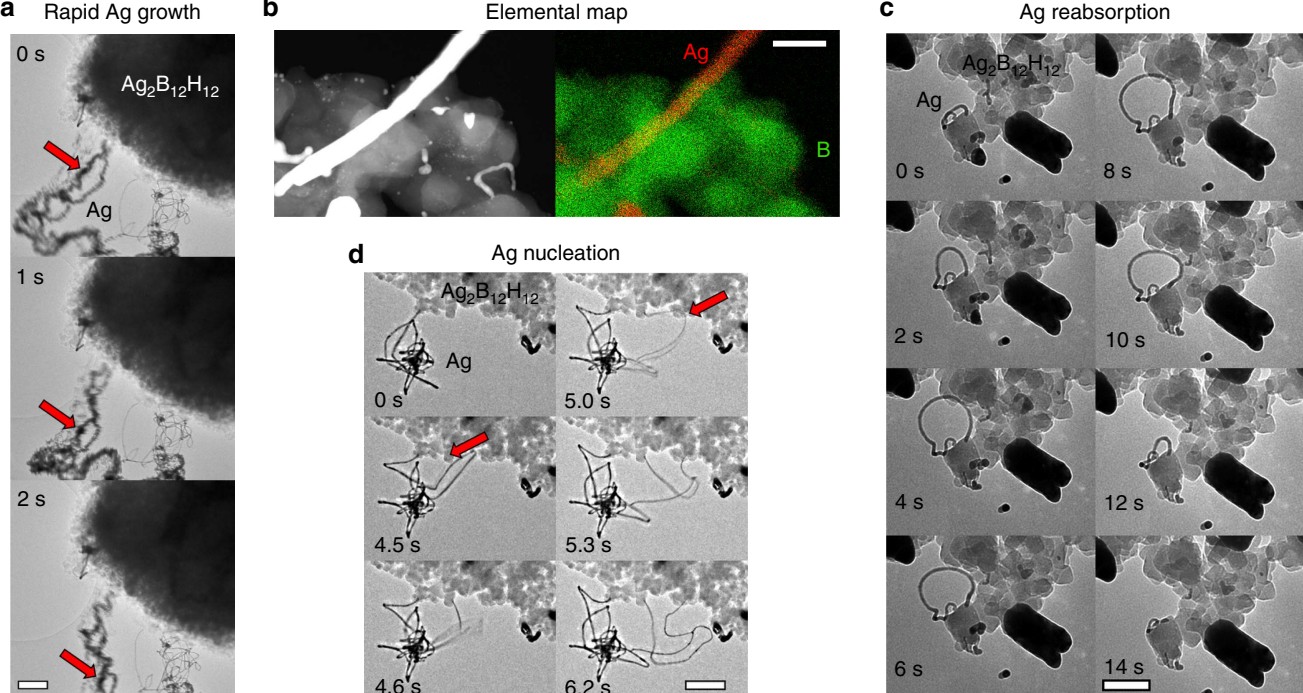

**Figure 3 | Silver nano-filament growth.** Transmission electron microscopy micrographs of Ag$_2$B$_{12}$H$_{12}$ demonstrating (**a**) fast Ag growth with an Ag particle moving as shown by the arrow (500 nm scale bar), (**b**) a dark field image and an elemental map showing silver (red) and boron (green) (100 nm scale bar), (**c**) Ag growth and reabsorption (250 nm scale bar) and (**d**) Ag nucleation points created where Ag filaments contact the surface of Ag$_2$B$_{12}$H$_{12}$ (see arrows) (100 nm scale bar). Image collection times are shown in seconds. Supplementary Movie 1 shows the dynamics of the process more clearly, collected in real time at 25 frames per second.

field, especially with regard to growth mechanisms under an electron beam[31].

The non-uniform electron beam can induce an electric potential difference across the $Ag_2B_{12}H_{12}$ sample, which could change polarity based on which parts of the particle are electron irradiated, stimulating $Ag^+$ migration to balance charge. This behaviour is similar to the controlled growth of Ag from a $Ag_2S$ scanning tunnelling microscope tip when a voltage is provided, and consequently Ag reabsorption when the voltage polarity is reversed[32]. Because of the electric potential difference in the TEM, we can describe the system shown in Fig. 3c as an $Ag|Ag_2B_{12}H_{12}|Ag$ battery, which can drive $Ag^+$ from one Ag site to another based on the electron reducing potential at each site. This ion transfer mechanism is the same as in batteries but is instead initiated by the electron beam. *In-situ* analysis of this rapid ion migration may provide new knowledge about electrochemical processes, however because the imaging mechanism is the same as the electron excitation mechanism it is difficult to control. In fact, this is a problem with other battery materials investigated by electron microscopy, where beam irradiation causes the material to undergo chemical and structural evolutions mirroring the charge–discharge cycle[33].

**Conclusions.** The silver boranes presented here show extreme silver mobility in the solid state and a step function in ion conductivity after an α-β polymorphic transition. The high temperature silver *closo*-borane structure reveals a correlation between ideal $Ag-H$ bond lengths and anion orientation. This demonstrates dynamic-anion facilitated $Ag^+$ migration, which can be described as a paddle-wheel mechanism for ion conduction if anion reorientation frequencies are found to match cation jump frequencies. These dynamics lead to high ion conductivities, which are enhanced two orders of magnitude at room temperature by forming new compounds with AgI. These materials exhibit the highest room temperature ion conductivities of any metal borane synthesized to date. Anion substitution may prove to be a powerful tool that can be used to tune the crystal structure of ion conductors, leading to order of magnitude increases in ion conductivity. There is still much to learn about the impact that different substituted anions have on the structure and ion conductivity of a material, especially in regard to structural dynamics.

Silver *closo*-boranes may also demonstrate a range of other useful properties. The 2.3 eV bandgap, in the middle of the visible spectrum, and the high stability of $Ag_2B_{10}H_{10}$ and $Ag_2B_{12}H_{12}$ may allow for the design of new photocatalysts in the future. These materials are also stable in water and could lead to new options for photocatalytic water splitting. These results also suggest that it would be promising to investigate the photoactive properties of other silver-based materials with weakly coordinating anions, which could also display useful bandgaps that may follow a trend with anion size. The silver *closo*-boranes display exceptionally fast and reversible silver growth from electron bombardment during TEM imaging on the nanoscale. Thus, the fast silver ion conductors presented here also provide significant insight into nano-growth mechanisms, which mimic battery charging and recharging processes.

## Methods

**Synthesis.** $(NH_4)_2B_{10}H_{10}$ was prepared by a known method from decaborane (Katchem) as follows:[10]

$$B_{10}H_{14} + 2(CH_3)_2S \rightarrow B_{10}H_{12} \cdot 2S(CH_3)_2 + H_2 \qquad (2)$$

$$B_{10}H_{12} \cdot 2S(CH_3)_2 + 2NH_3 \rightarrow (NH_4)_2B_{10}H_{10} + 2(CH_3)_2S \qquad (3)$$

The product was dried under vacuum at 70 °C to remove excess dimethylsulfide and ammonia.

$Ag_2B_{10}H_{10}$ was prepared by adding an aqueous 1 M $AgNO_3$ solution dropwise to $(H_3O)_2B_{10}H_{10}$ in milli-Q water in a 2:1 molar ratio. The acid, $(H_3O)_2B_{10}H_{10}$, was prepared by passing an aqueous solution of $(NH_4)_2B_{10}H_{10}$ through an amberlite IR120-H ion exchange resin (Fluka). The solution was stirred and an off-white coloured precipitate, $Ag_2B_{10}H_{10}$, was immediately formed as $AgNO_3$ was added. The precipitate quickly darkened in colour over the course of minutes to dark brown as the result of light exposure. The suspension was then filtered and heated to 70 °C under vacuum. The characteristic $B-H$ bending modes of $B_{10}H_{10}^{2-}$ (2,550 and 2,350 cm$^{-1}$) are clearly observed by FTIR (Supplementary Fig. 13).

$Ag_2B_{12}H_{12}$ was prepared by adding an aqueous 1 M $AgNO_3$ solution dropwise to $(H_3O)_2B_{12}H_{12}$ in milli-Q water in a 2:1 molar ratio. The acid, $(H_3O)_2B_{12}H_{12}$, was prepared by passing an aqueous solution of $Li_2B_{12}H_{12}$ (Katchem) through an amberlite IR120-H ion exchange resin (Fluka). The solution was stirred and an off-white coloured precipitate, $Ag_2B_{12}H_{12}$, was immediately formed as $AgNO_3$ was added. The suspension was then filtered and heated to 70 °C under vacuum. The characteristic $B-H$ bending modes of $B_{12}H_{12}^{2-}$ (2,450 and 2,320 cm$^{-1}$) are clearly observed by FTIR (Supplementary Fig. 13).

These new silver *closo*-boranes are insoluble in water and can be handled both in water or air without hydrate formation or decomposition, unlike most other metal boranes, which are often hygroscopic or reactive[6,34].

$AgI-Ag_2B_{10}H_{10}$ was hand ground in a 1:1 molar ratio of AgI (Alfa Aesar) and $Ag_2B_{10}H_{10}$ followed by heat treatment under argon to 200 °C for 1 h. *In-situ* SR-PXD data of the formation of $Ag_{(2+x)}I_xB_{10}H_{10}$ (possibly with $x\sim1$) during constant heating is shown in Supplementary Fig. 8. A discussion of the crystal structure determination is provided in Supplementary Note 1.

$AgI-Ag_2B_{12}H_{12}$ was hand ground in a 1:1 molar ratio of AgI and $Ag_2B_{10}H_{10}$ followed by heat treatment under argon to 250 °C for 1 h. *In-situ* SR-PXD data of formation of $Ag_{(2+x)}I_xB_{12}H_{12}$ (possibly with $x\sim1$) during constant heating is shown in Supplementary Fig. 9. A discussion of the crystal structure determination is provided in Supplementary Note 1.

All chemicals and samples were stored in the absence of light and only handled briefly in light during synthesis and characterization.

**Characterization.** *In-situ* synchrotron radiation powder X-ray diffraction (SR-PXD) data were collected at the I711 beamline at MAX II, MAX-lab, Lund, Sweden with $\lambda = 0.9938$ Å on a Titan CCD 165 mm detector. A hot air blower was used to heat samples at 5 °C per min under argon in a sapphire capillary using a custom made sample cell[35]. SR-PXD data for structure solution were obtained from the P02.1 beamline at Petra III, DESY, Hamburg, Germany at $\lambda = 0.20775$ Å on a PerkinElmer XRD1621CN3-EHS 410 mm detector. Structure solutions were performed in free objects for crystallography (FOX) and FullProf using rigid body $B_{10}H_{10}^{2-}$ and $B_{12}H_{12}^{2-}$ anions.

Electrochemical Impedance Spectroscopy data were collected on a BioLogic MTZ-35 impedance analyser equipped with a high temperature sample holder. Samples were pressed into 6.35 mm diameter pellets of ca. 1 mm thickness and 100 μm gold foil was mechanically fixed to both sides of the pellet, before being placed between platinum electrodes. All measurements were conducted in an argon atmosphere and temperature was measured by a K-type thermocouple 5 mm from the sample. Impedance data were measured at 100 mV ac from 1 to $1 \times 10^7$ Hz. Ion conductivity data ($\sigma$) were derived from Nyquist impedance plots (Supplementary Fig. 14) using the $x$-intercept of the Nyquist semicircle ($I$), area of the pellet face ($A$) and pellet thickness ($t$) according to: $\sigma = t/(I \times A)$.

Transmission electron microscopy (TEM) micrographs, Scanning TEM (STEM), and energy dispersive spectroscopy data acquisition was performed on an FEI Talos system equipped with a 200 kV FEG. The STEM/energy dispersive spectroscopy data were collected using the ChemiSTEM system with a High Angle Annular Dark Field (HAADF) detector setup. Rapid TEM micrograph collection was undertaken at 25 frames per second, while collecting 512 × 512 pixel images. Samples were prepared by suspension in toluene and ultrasonication before being dropped onto a holey carbon covered copper grid. The TEM results were reproducible in multiple $Ag_2B_{12}H_{12}$ samples prepared in different batches. $Ag_2B_{12}H_{12}$ exhibited more active Ag growth than $Ag_2B_{10}H_{10}$ and the Ag growth activity was also higher in freshly prepared samples (<4 weeks).

Ultraviolet/VIS data were collected on a Shimadzu ultraviolet-3600 spectrophotometer from 220 to 1,500 nm in 0.5 nm steps. Powders were pressed onto a flat plate $BaSO_4$ background in air before analysis. The direct bandgap was calculated using the Kubelka–Munk function with a Tauc plot from $(F(R)h\nu)^2 = A(h\nu - E_g)$, where $h\nu = 1239.7/\lambda$ eV, where $\lambda$ is the wavelength in nm, $F(R)$ is the reflectance spectrum, $A$ is a proportional constant and $E_g$ is the band gap.

FTIR spectra were collected on a NICOLET 380 FT-IR (Thermo Scientific) coupled to a Smart Orbit stage for attenuated total reflectance analysis. The spectra were collected in the wavenumber range of 400–4,000 cm$^{-1}$ with 32 scans.

Differential scanning calorimetry was conducted on a PerkinElmer STA 6,000, where 15–20 mg samples were placed in Al crucibles and heated at 5 °C min$^{-1}$ under constant argon flow (20 ml min$^{-1}$). The instrument was coupled to a Hiden Analytical HPR-20 quadrupole mass spectrometer for residual gas analysis.

**Data availability.** The data that support the findings of this study are available from the corresponding author on reasonable request. The X-ray crystallographic

coordinates for structures reported in this study have been deposited at the Cambridge Crystallographic Data Centre (CCDC), under deposition numbers CSD-431821—CSD-431824. These data can be obtained free of charge from The Cambridge Crystallographic Data Centre via http://www.ccdc.cam.ac.uk/data_request/cif.

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

## Acknowledgements

M.P. acknowledges financial support from The Danish Council for Independent Research for DFF Mobility 1325-00072. We are grateful to the Carlsberg Foundation and Danish Council for Independent Research, DFF 4181-00462 (HyNanoBorN), The Innovation Fund Denmark (HyFill-Fast), The European Marie Curie Actions under ECOSTORE grant agreement n° 607040 and the Danish National Research Foundation, Center for Materials Crystallography (DNRF93). Parts of this research were carried out at the light source Petra III at DESY, a member of the Helmholtz Association (HGF), and at beamline I711, in the research laboratory MAXIV, MAX II synchrotron, Lund, Sweden. We are grateful to the Villum Foundation for funding the FEI Talos TEM.

## Author contributions

M.P. and B.R.S.H. conducted material synthesis and characterization. M.J. and M.P. undertook crystallographic structure solutions. B.R. undertook TEM investigations. T.R.J. managed the project and aided in writing the manuscript with M.P., B.R.S.H. and M.J.

## Additional information

**Competing interests:** The authors declare no competing financial interests.

**Publisher's note**: 

