## [Peer Review File · Nature Communications]

Reviewers' comments:

Reviewer #1 (Remarks to the Author):

The manuscript describes the multifunctions of newly synthesized silver-based hydrides Ag₂B₁₀H₁₀, Ag₂B₁₂H₁₂ and the derivatives, which touches the cutting edge topic on designing of chemical hydrides with novel functions. The experimental results indicate that Ag₂B₁₀H₁₀ and Ag₂B₁₂H₁₂ are semiconductors with a bandgap at 2.3 eV in the green visible light spectrum, displaying photosensitivity analogous to silver halides. The ion conductivity of Ag₂B₁₀H₁₀ and Ag₂B₁₂H₁₂ can be enhanced more than two orders of magnitude at room temperature (up to 3.2 mS/cm) by anion substitution with AgI. More interestingly, it was found that fast silver nano-filament growth when excited by electrons during transmission electron microscope investigations. These experimental results with high novelty strongly support the authors' conclusions and provide wide interest for several scientific fields like photo-catalyst, solid state ionics, nanowire and hydrides. Therefore, this manuscript can be acceptable for publication with minor revisions.

1. Please increase the resolution of Fig. 2.

2. Please revise the reference in the required style, e.g. ref. 11.

Reviewer #2 (Remarks to the Author):

The manuscript presents the characterisation of two silver closo-boranes and two iodised closo-boranes including their ionic conductivity and thermal stability. In addition it is shown that the compounds are semiconductors and can release and re-absorb metallic silver in the form of nano-fibers when exposed to the electron beam.

The results are of general interest, because the closo-boranes of transition metals are rare due to the difficulty to stabilize the given oxidation state of metals when exposed to the reducing environment of the borane.

The main claims of the paper are novel application of the closo-boranes, i.e. semiconductors and metallic nanofibers. While referee agrees with both, another claim of the manuscript, a direct proof of a paddle-wheel mechanism of ionic conductivity seems not to be justified.

The manuscript is promising for the publication in Nature Comm., but needs modification and additional experiments.

In particular:

- Information of known closo-boranes of transition metals is missing in the introduction.
- The crystal structures of Ag₃IB₁₂H₁₂ and Ag₃IB₁₀H₁₀ are not presented in the manuscript contrary to what the authors claim. Their powder diffraction data presented in figures S12 and S13 seem to allow the structural characterization.
- The B₁₂H₁₂²⁻ anions cannot be localized on fcc lattice if the space group is Pa-3.
- The proof of paddle-wheel mechanism as given in the manuscript is not satisfactory. The refined disordered structure of the HT-Ag₂B₁₂H₁₂ really corresponds to the two orientations of the closo-anion as given in the figure S5, but it is impossible to distinguish from the Bragg signals in the diffraction pattern whether this orientation disorder is of static or dynamic origin. No correlation with the Ag⁺ cation movement is therefore possible. BTW the disorder seems to be different than in the HT-Li₂B₁₂H₁₂, and these two structures are not strictly iso-structural.
- The existence of Ag₃IB₁₂H₁₂ and Ag₃IB₁₀H₁₀ is not proved from the experimental data. The chemical reaction between AgI and Ag-closo-borane does not produce a single product, but either metallic silver or AgI are still visible in the powder patterns. Without a structural model of these two new compounds with high ionic conductivity the discussion of them should be more limited.
- The very interesting result is the extraction and re-insertion of the metallic silver from and to the silver closo-borane. The supplied supporting material is convincing. It would be nice to prove the

reduction and oxidation of the silver by an NEXAFS experiment in the in-situ electrochemical cell.

In conclusion: in its actual form the manuscript is not ready for the publication in Nature Comm. Additional effort and experiments are needed to better characterize the two high conducting phases and to explain the mechanism of extraction and re-insertion of the silver.

Reviewer #3 (Remarks to the Author):

. This paper reported $\text{Ag}_2\text{B}_{10}\text{H}_{10}$ and $\text{Ag}_2\text{B}_{12}\text{H}_{12}$ and the iodo-derivatives $\text{Ag}_3\text{IB}_{10}\text{H}_{10}$ and $\text{Ag}_3\text{IB}_{12}\text{H}_{12}$ as solid-state electrolytes. The mechanism of silver conductivity is also investigated, and is consistent with the well-established paddle-wheel mechanism. The materials display an impressive Ag^+ conductivity as well as good stability. The systematical study of the phase transition and structure dynamics could provide insights for the design of solid electrolytes with a higher conductivity. The following issues should be addressed before it could be published.

1. The organization of the MS needs to be improved. Several interesting properties of silver closo-borates are reported, however, the data is sometimes presented without proper context. The bandgap of $\text{Ag}_2\text{B}_{10}\text{H}_{10}$ and $\text{Ag}_2\text{B}_{12}\text{H}_{12}$ is reported, but it is unclear what the relevance of these measurements is.
2. Similarly, the Ag nanostructures are very interesting, but it is unclear how much control over size and morphology can be achieved.
3. The manuscript lacks a true conclusion section. It would be helpful to provide some more details about the lessons learnt, and perhaps some design rules for highly conductive solid electrolytes.

The authors would like to thank the Reviewers for their insightful and useful comments. Our responses to individual queries are below.

Reviewers' comments:

Reviewer #1 (Remarks to the Author):

The manuscript describes the multifunctions of newly synthesized silver-based hydrides Ag₂B₁₀H₁₀, Ag₂B₁₂H₁₂ and the derivatives, which touches the cutting edge topic on designing of chemical hydrides with novel functions. The experimental results indicate that Ag₂B₁₀H₁₀ and Ag₂B₁₂H₁₂ are semiconductors with a bandgap at 2.3 eV in the green visible light spectrum, displaying photosensitivity analogous to silver halides. The ion conductivity of Ag₂B₁₀H₁₀ and Ag₂B₁₂H₁₂ can be enhanced more than two orders of magnitude at room temperature (up to 3.2 mS/cm) by anion substitution with AgI. More interestingly, it was found that fast silver nano-filament growth when excited by electrons during transmission electron microscope investigations. These experimental results with high novelty strongly support the authors' conclusions and provide wide interest for several scientific fields like photo-catalyst, solid state ionics, nanowire and hydrides. Therefore, this manuscript can be acceptable for publication with minor revisions.

1. Please increase the resolution of Fig. 2.

Response: Figure 2 appears low resolution in the pdf build of the manuscript, however, a high resolution version of the figure will be provided with the final submission.

2. Please revise the reference in the required style, e.g. ref. 11.

Response: The reference list has now been updated to the required style.

Reviewer #2 (Remarks to the Author):

The manuscript presents the characterisation of two silver closo-boranes and two iodised closo-boranes including their ionic conductivity and thermal stability. In addition it is shown that the compounds are semiconductors and can release and re-absorb metallic silver in the form of nano-fibers when exposed to the electron beam.

The results are of general interest, because the closo-boranes of transition metals are rare due to the difficulty to stabilize the given oxidation state of metals when exposed to the reducing environment of the borane.

The main claims of the paper are novel application of the closo-boranes, i.e. semiconductors and metallic nanofibers. While referee agrees with both, another claim of the manuscript, a direct proof

of a paddle-wheel mechanism of ionic conductivity seems not to be justified.

The manuscript is promising for the publication in Nature Comm., but needs modification and additional experiments.

In particular:

- Information of known closo-boranes of transition metals is missing in the introduction.

Response: A few sentences have now been added to the introduction to describe other known transition metal closo-boranes, as follows:

"A large variety of transition metal closo-boranes have previously been synthesised (i.e. Ag, Cd, Co, Cr, Cu, Fe, Hg, Mn, Ni, Pd, Sc, Zn),⁶ in most cases as solvates due to the strong solvent coordination to the cation. In contrast, the silver cation has a lower charge density and can be isolated as a solvent-free closo-borane,¹⁰ but interestingly, only the crystal structures of silver *closo*-borane solvates have previously been described.^{11"}

- The crystal structures of Ag₃IB₁₂H₁₂ and Ag₃IB₁₀H₁₀ are not presented in the manuscript contrary to what the authors claim. Their powder diffraction data presented in figures S12 and S13 seem to allow the structural characterization.

Response: We do not claim to have solved the crystal structures of Ag₃IB₁₂H₁₂ and Ag₃IB₁₀H₁₀. We have clarified parts of the text to make this clear. The data in Figures S12 and S13 do demonstrate the formation of a new compound (correlated with the consumption of most of the reagents, AgI and Ag₂B_xH_x). However, the quality of the data is not suitable for structure solution.

There are a number of issues that prevent structure solution. We have made new samples and collected new diffraction data on these compounds in the laboratory and at 2 different synchrotron facilities but are still unable to index or solve the crystal structures with confidence. We have now added the following information to the supporting information, which will be helpful to other researchers in the future.

- Firstly, the new compounds, Ag₃IB₁₂H₁₂ and Ag₃IB₁₀H₁₀, have a small crystallite size and only exhibit 9 diffraction peaks over a q -range of $0.2 - 8 \text{ \AA}^{-1}$.
- Secondly, there are traces of reagents or decomposition products in the diffraction patterns that may overlap with minor peaks, thus making space group indexing particularly challenging in this case.
- It is also possible that the crystal structures of these highly conducting materials exhibit structural dynamics (i.e. cation partial occupancy and/or anion reorientation), which require high quality diffraction data, ideally on a single-phase compound.
- There is also the possibility that AgI-based compounds could provide misleading diffracted intensities. This phenomenon has been observed in AgI powders and has been attributed to primary extinctions from a lack of mosaicity [Burley, G. Photolytic behavior of silver iodide. J. Res. Natl Bur. Stand. 67A, 301–307 (1963)].
- Our preliminary research indicates that the unit cell of the new compounds, Ag₃IB₁₂H₁₂ and Ag₃IB₁₀H₁₀, could be large $\sim 2400 \text{ \AA}^3$. Unfortunately, with so few diffraction peaks and possible peak overlap with impurities, indexing such a large unit cell is currently unreliable.

We believe that our data (Figures S12 and S13) do demonstrate the formation of new compounds with stoichiometry $\text{Ag}_3\text{IB}_{12}\text{H}_{12}$ and $\text{Ag}_3\text{IB}_{10}\text{H}_{10}$. These compounds demonstrate excellent room temperature ion conductivities, but their structural characterisation is particularly challenging. It should be noted that the structural solution of the high-temperature polymorph of AgI required an overwhelming number of challenging structural studies over a 60 year period in order to develop an accurate structural model. [S. Hull, Rep. Prog. Phys., 2004, 67, 1233.] We believe that an accurate structural model for $\text{Ag}_3\text{IB}_{12}\text{H}_{12}$ and $\text{Ag}_3\text{IB}_{10}\text{H}_{10}$ will also require challenging X-ray and neutron diffraction studies.

- The B12H12⁻ anions cannot be localized on fcc lattice if the space group is Pa-3.

Response: This is not true. The anions are localised on an fcc lattice in a simple cubic unit cell. The same phenomenon occurs for C_{60} , which is described in detail in [Harris, A., & Sachidanandam, R. (1992). Orientational Ordering of Icosahedra in Solid C60. Physical Review B, 46 (8), 4944-4957].

- The proof of paddle-wheel mechanism as given in the manuscript is not satisfactory. The refined disordered structure of the HT-Ag₂B₁₂H₁₂ really corresponds to the two orientations of the closo-anion as given in the figure S5, but it is impossible to distinguish from the Bragg signals in the diffraction pattern whether this orientation disorder is of static or dynamic origin. No correlation with the Ag⁺ cation movement is therefore possible. BTW the disorder seems to be different than in the HT-Li₂B₁₂H₁₂, and these two structures are not strictly iso-structural.

Response: We have expanded our discussion of the proposed paddle-wheel mechanism. We now also relate our finding to earlier studies that prove dynamic disorder in other *closo*-borane systems as follows:

“The closo-borane anions are structurally described by reoriented partially occupied B-H polyhedra. The structural model extracted from diffraction data cannot differentiate between static or dynamic orientational disorder, but dynamic disorder is well known in other borane systems from comprehensive studies using nuclear magnetic resonance (NMR), quasielastic neutron scattering (QENS), and neutron vibrational spectroscopy (NVS).^{16, 17, 18}”

It is very important to note that the anion orientations correlate with Ag-H bond lengths. For particular anion orientations some Ag crystallographic sites are physically unrealistic, suggesting that the Ag must occupy other sites when the anion is in these orientations, thus supporting the theory of a “paddle-wheel” mechanism. We have now further clarified this in the manuscript and supporting information by adding in a new figure (S5) and adding further discussion:

“The silver ions in the β -structures are distributed over two different sites, $32f$ and $8c$, occupying each crystallographic site equally. The $8c$ site is tetrahedrally surrounded by four $32f$ sites, as shown by the transparent tetrahedra in Figure 1, but only one of these five sites can be occupied by Ag simultaneously ($32f$ site separation ~ 2.5 Å compared to the Ag^+ diameter of 2.58 Å). Each face of the tetrahedron is directed towards an anion superposition and the anions generate an inverted tetrahedron with respect to the Ag-tetrahedron around the $8c$ Wyckoff site. The $8c$ Ag sites permit plausible Ag-H bond lengths (2.0 – 2.6 Å) for all anion orientations, but are more likely in certain orientations (see Figure S5). The $32f$ Ag sites are subjected to a large variability in Ag-H bond lengths

(1.4 – 2.8 Å) depending on the anion orientation. In many cases the 32f Ag site has an Ag–H bond length that is physically unrealistic when compared to known Ag–H bond lengths (1.8 – 2.7 Å).¹¹ This suggests that Ag must occupy other crystallographic sites when the anion is in these orientations, thus supporting the theory of a ‘paddle-wheel’ mechanism for Ag⁺ migration, where Ag-ions may be forced to jump from site-to-site by rotating anions.”

It is true that high temperature β -Li₂B₁₂H₁₂ and β -Ag₂B₁₂H₁₂ are not isostructural and we do not state this. We mention that the room temperature α -Li₂B₁₂H₁₂ and room temperature α -Ag₂B₁₂H₁₂ are isostructural. This is now been clarified in the text.

- The existence of Ag₃IB₁₂H₁₂ and Ag₃IB₁₀H₁₀ is not proved from the experimental data. The chemical reaction between AgI and Ag-closo-borane does not produce a single product, but either metallic silver or AgI are still visible in the powder patterns. Without a structural model of these two new compounds with high ionic conductivity the discussion of them should be more limited.

Response: This is not correct. The *in-situ* diffraction data Figure S12 and S13 clearly show decreasing diffracted intensity from AgI and formation of a new compound with a fixed stoichiometry, i.e. not a solid solution. We also observe the reformation of AgI after the decomposition of Ag₃IB₁₂H₁₂, thus proving its incorporation in the new compound. However, we agree that the chemical reaction is not complete under the presented conditions, i.e. constant heating.

We also note that compounds of the same stoichiometry are also known, which we now discuss and reference: “Other 1:1 stoichiometric compounds incorporating metal iodides and boranes have been shown previously for M₃IB₁₂H₁₂ (M = NH₄, K, Rb, Cs).¹⁹”

We only discuss Ag₃IB₁₂H₁₂ and Ag₃IB₁₀H₁₀ in regard to their ion conductivity, which is valid no matter what structure they have. In fact, these results also demonstrate that the impressive ion conductivity arises from new compounds and not their reagents. The ion conductivities of Ag₃IB_xH_x are higher than Ag₂B_xH_x, but also lower than the high temperature AgI, thus showing that the conductivity does not arise from either starting reagent, but instead from a new compound.

- The very interesting result is the extraction and re-insertion of the metallic silver from and to the silver closo-borane. The supplied supporting material is convincing. It would be nice to prove the reduction and oxidation of the silver by an NEXAFS experiment in the *in-situ* electrochemical cell.

Response: We agree that the supporting information is convincing, especially the TEM movie. We have also shown that the nano-filaments are indeed Ag metal from elemental maps, mirroring the crystalline Ag observed after thermal decomposition. Although we believe that *in-situ* NEXAFS experiments would be interesting, we suggest that they are most appropriate for further studies as they are quite specialised and may not add so much detail to the fundamental understanding of the system.

In conclusion: in its actual form the manuscript is not ready for the publication in Nature Comm. Additional effort and experiments are needed to better characterize the two high conducting phases and to explain the mechanism of extraction and re-insertion of the silver.

Response: We believe that the mechanism for Ag extraction and re-insertion is well described by the reduction reaction: $\text{Ag}^+ + \text{e}^- \leftarrow \rightarrow \text{Ag}$, now included in the manuscript, along with the measurements of fast Ag^+ ion conductivity. We have now added several sentences to elaborate further on this interesting process:

“The Ag formation mechanism may begin in a similar way to the photographic process, where silver atoms are formed at a point within a silver salt when illuminated by photons. These photons promote the formation of electron hole pairs via the photovoltaic effect and a free electron can encounter a lattice defect causing it to become negatively charged, capture Ag^+ ions, and form Ag.³³”

and

“There is limited control over the growth process within the TEM, but an increase in electron current typically initiates or enhances Ag growth, which is still localised to particular nucleation sites. Interestingly, the Ag morphology is always filament-like from these nucleation sites, where silver is drawn from distant regions of the powder. However, the electron-driven process is different to thermal decomposition where thermally treated $\text{Ag}_2\text{B}_{12}\text{H}_{12}$ shows 50 - 500 nm Ag particles without filament growth (Figure S10). The study of nanocrystal growth mechanisms is still a developing field, especially with regard to growth mechanisms under an electron beam.³⁴”

Further research into the crystal structures of $\text{Ag}_3\text{IB}_{12}\text{H}_{12}$ and $\text{Ag}_3\text{IB}_{10}\text{H}_{10}$ is worthy of study but cannot yet be achieved (see response 2 to reviewer 2).

Reviewer #3 (Remarks to the Author):

This paper reported $\text{Ag}_2\text{B}_{10}\text{H}_{10}$ and $\text{Ag}_2\text{B}_{12}\text{H}_{12}$ and the iodo-derivatives $\text{Ag}_3\text{IB}_{10}\text{H}_{10}$ and $\text{Ag}_3\text{IB}_{12}\text{H}_{12}$ as solid-state electrolytes. The mechanism of silver conductivity is also investigated, and is consistent with the well-established paddle-wheel mechanism. The materials display an impressive Ag^+ conductivity as well as good stability. The systematical study of the phase transition and structure dynamics could provide insights for the design of solid electrolytes with a higher conductivity. The following issues should be addressed before it could be published.

1. The organization of the MS needs to be improved. Several interesting properties of silver closo-borates are reported, however, the data is sometimes presented without proper context. The bandgap of $\text{Ag}_2\text{B}_{10}\text{H}_{10}$ and $\text{Ag}_2\text{B}_{12}\text{H}_{12}$ is reported, but it is unclear what the relevance of these measurements is.

Response: Some extra sentences have been added throughout the manuscript to elaborate on results. The bandgap is now discussed in the proper context as follows:

“A series of Ag-based photocatalysts were recently identified,²⁸ with Ag_3PO_4 demonstrating a quantum yield of nearly 90% (bandgap ~ 2.4 eV), performing well in both water splitting and wastewater cleaning applications.²⁹ Thus, the silver closo-boranes may also show promise as future photocatalysts due to their water stability and favourable bandgap.”

2. Similarly, the Ag nanostructures are very interesting, but it is unclear how much control over size

and morphology can be achieved.

Response: This is an interesting point. We have already noted that “the imaging mechanism is the same as the electron excitation mechanism thus is difficult to control.” However, this is a complex process that should be studied in more detail in the future. We have now added some information regarding the level of control we have during TEM imaging:

“There is limited control over the growth process within the TEM, but an increase in electron current typically initiates or enhances Ag growth, which is still localised to particular nucleation sites. Interestingly, the Ag morphology is always filament-like from these nucleation sites, where silver is drawn from distant regions of the powder. However, the electron-driven process is different to thermal decomposition where thermally treated $\text{Ag}_2\text{B}_{12}\text{H}_{12}$ shows 50 - 500 nm Ag particles without filament growth (Figure S10). The study of nanocrystal growth mechanisms is still a developing field, especially with regard to growth mechanisms under an electron beam.³⁴”

3. The manuscript lacks a true conclusion section. It would be helpful to provide some more details about the lessons learnt, and perhaps some design rules for highly conductive solid electrolytes.

Response: We have now extended the conclusion to suggest some future research avenues based on the new results presented in the manuscript:

“Anion substitution may prove to be a powerful tool that can be used to tune the crystal structure of ion conductors, leading to order of magnitude increases in ion conductivity. There is still much to learn about the impact that different substituted anions have on the structure and ion conductivity of a material, especially in regard to its structural dynamics.”

“These results also suggest that it would be promising to investigate the photoactive properties of other silver-based weakly coordinating anions, which could also display useful bandgaps that may follow a trend with anion size.”

Reviewers' comments:

Reviewer #1 (Remarks to the Author):

The authors have modified the manuscript according to the comments of all referees, but some important comments have not been taken into account:

- The unknown highly conducting compounds should not be called Ag₃IB₁₂H₁₂ and Ag₃IB₁₀H₁₀ as there are no proofs about their chemical composition. The compounds are certainly formed by the reaction between AgI and Ag₂B₁₂H₁₂ and Ag₂B₁₀H₁₀, resp., but the ratio may be (and probably is) different from 1:1. The authors correctly estimate that their structure may be more complex, i.e. more complex than the structures of alkali metal halides-closoboranes, another indicator that the ratio is not 1:1.
- The primary extinction effect cannot be strong in powders with very small grains.
- The B₁₂H₁₂ anion cannot be localized on fcc lattice in the space group Pa-3, because there is no fcc lattice associated to that group. The anion can be localized on fcc sublattice. The comment was, however, not only related to the terminology. The referee guesses that the two anions (one in the corner of the cubic cell, and the other in the center of the cubic faces) do not have the same orientation, and do not form therefore fcc sublattice.
- The modified discussion of the structural results still do not prove the paddle-wheel mechanism in beta-Ag₂B₁₂H₁₂: There are four different orientations of the B₁₂H₁₂ anion, but only two are discussed. If the cation jumps are somehow correlated to anion reorientation, the probability of various orientations (i.e. occupancy factors) should be correlated to occupation of the two cation sites. Moreover, the paddle-wheel mechanism is reserved to the case when one anion jump promotes one cation move, i.e. the dynamics of both, cations and anions, is on the same frequency scale. And this is not possible to prove from Bragg diffraction data. There is without no doubt an effect of anions dynamics on cations jumps, but the mechanism is unclear. The term paddle-wheel should be therefore removed.

The manuscript cannot be accepted for the publication unless the above mentioned comments are addressed.

Reviewer #2 (Remarks to the Author):

The manuscript describes the conductivity and unique ionic transport properties of silver-based close-borates Ag₂B₁₀H₁₀, Ag₂B₁₂H₁₂ and the dodo-derivatives. The revised version of the manuscript is greatly improved, therefore, the manuscript is now acceptable for publication.

NCOMMS-16-23746A

Multifunctionality of Silver closo-Boranes

Mark Paskevicius, Bjarne R. S. Hansen, Mathias Jørgensen, Bo Richter, Torben R. Jensen

We thank the Reviewers for their dedication and positive comments. We have now clarified these issues as discussed below.

Reviewers' comments:

Reviewer #1 (Remarks to the Author):

The authors have modified the manuscript according to the comments of all referees, but some important comments have not been taken into account:

- The unknown highly conducting compounds should not be called Ag₃IB₁₂H₁₂ and Ag₃IB₁₀H₁₀ as there are no proofs about their chemical composition. The compounds are certainly formed by the reaction between AgI and Ag₂B₁₂H₁₂ and Ag₂B₁₀H₁₀, resp., but the ratio may be (and probably is) different from 1:1. The authors correctly estimate that their structure may be more complex, i.e. more complex than the structures of alkali metal halides-closoboranes, another indicator that the ratio is not 1:1.

We have now renamed these compounds to Ag_(2+x)I_xB₁₀H₁₀ and Ag_(2+x)I_xB₁₂H₁₂ to allow for the possibility that the exact composition, $x = 1$, does not form.

Furthermore, the text on page 5 is rewritten as:

“Two new compounds were discovered in AgI–Ag₂B₁₀H₁₀ and AgI–Ag₂B₁₂H₁₂ systems, synthesised by annealing 1:1 compositions to study the influence of iodide substitution on ion conductivity. *In situ* SR-PXD of these composites reveals discontinuous changes in Bragg diffraction (see Figure S12 and S13), which suggest a fixed stoichiometry for new compounds, denoted Ag_(2+x)I_xB₁₀H₁₀ and Ag_(2+x)I_xB₁₂H₁₂, possibly with $x \sim 1$, rather than the formation of solid solutions.”

- The primary extinction effect cannot be strong in powders with very small grains.

This is true and we do not mention this in the manuscript. The mention of this phenomenon has been removed from the supporting information.

- The B₁₂H₁₂ anion cannot be localized on fcc lattice in the space group Pa-3, because there is no fcc lattice associated to that group. The anion can be localized on fcc sublattice. The comment was, however, not only related to the terminology. The referee guess that the two anions (one in the corner of the cubic cell, and the other in the center of the cubic faces) do not have the same orientation, and do not form therefore fcc sublattice.

We have now removed the mention of an fcc lattice and instead refer to the anion arrangement as follows: “The $B_{12}H_{12}^{2-}$ anions are centred on the faces and corners of the unit cell while Ag^+ occupies the 8c Wyckoff sites.”

- The modified discussion of the structural results still do not prove the paddle-wheel mechanism in beta- $Ag_2B_{12}H_{12}$: There are four different orientations of the $B_{12}H_{12}$ anion, but only two are discussed. If the cation jumps are somehow correlated to anion reorientation, the probability of various orientations (i.e. occupancy factors) should be correlated to occupation of the two cation sites. Moreover, the paddle-wheel mechanism is reserved to the case when one anion jump promote one cation move, i.e. the dynamics of both, cations and anions, is on the same frequency scale. And this is not possible to prove from Bragg diffraction data. There is without no doubt an effect of anions dynamics on cations jumps, but the mechanism is unclear. The term paddle-wheel should be therefore removed.

We have now removed the term ‘paddle-wheel’. As the reviewer suggests there is no doubt that there is an effect on cation jumps based on anion dynamics. As such, we refer to this process as “dynamic-anion facilitated Ag^+ migration”. We agree with the reviewer that cation-anion dynamics must occur on the same frequency scale for the process to be denoted “paddle-wheel” and we now clarify this issue in the discussion as follows: “This demonstrates dynamic-anion facilitated Ag^+ migration, which can be described as a paddle-wheel mechanism for ion conduction if anion reorientation frequencies are found to match cation jump frequencies. ”

The manuscript cannot be accepted for the publication unless the above mentioned comments are addressed.

We have now addressed all comments.

Reviewer #2 (Remarks to the Author):

The manuscript describes the conductivity and unique ionic transport properties of silver-based close-borates $Ag_2B_{10}H_{10}$, $Ag_2B_{12}H_{12}$ and the dodo-derivatives. The revised version of the manuscript is greatly improved, therefore, the manuscript is now acceptable for publication.

Reviewers' comments:

Reviewer #1 (Remarks to the Author):

The authors have correctly addressed all comments. The manuscript is ready for being accepted to Nature Communications.